# Untargeted Plasma Metabolomics Extends the Biomarker Profile of Mitochondrial Neurogastrointestinal Encephalomyopathy

**DOI:** 10.3390/ijms26189107

**Published:** 2025-09-18

**Authors:** Bridget E. Bax, Sema Kalkan Uçar

**Affiliations:** 1School of Health and Medical Sciences, City St George’s, University of London, London SW17 0RE, UK; 2Division of Metabolism and Nutrition, Department of Pediatrics, Ege University Medical Faculty, Izmir 35100, Turkey; sema.kalkan.ucar@ege.edu.tr

**Keywords:** MNGIE, mitochondrial neurogastrointestinal encephalomyopathy, metabolomics, *TYMP*, untargeted metabolomic profiling, biomarkers

## Abstract

Mitochondrial neurogastrointestinal encephalomyopathy (MNGIE) is caused by pathogenic mutations in the nuclear *TYMP* gene, which encodes the cytosolic enzyme thymidine phosphorylase. In addition to the systemic accumulation of thymidine and deoxyuridine, several case studies have reported abnormalities in a range of other metabolites in patients with MNGIE. Since metabolites are intermediates or end-products of numerous biochemical reactions, they serve as highly informative indicators of an organism’s metabolic activity. This study aimed to perform an untargeted metabolomic profiling to determine whether individuals with MNGIE exhibit a distinct plasma metabolic signature compared to 15 age- and sex-matched healthy controls. Metabolites were profiled using Ultra-High-Performance Liquid Chromatography–Mass Spectrometry (UHPLC-MS). A total of 160 metabolites were found to be significantly upregulated and 260 downregulated in patients with MNGIE. KEGG pathway enrichment analysis revealed disruptions in 20 metabolic pathways, with arachidonic acid metabolism and bile acid biosynthesis being the most significantly upregulated. Univariate receiver operating characteristic (ROC) curve analyses identified 23 individual metabolites with diagnostic potential, each showing an area under the curve (AUC) ≥ 0.80. We propose that an impaired resolution of inflammation contributes to a chronic inflammatory state in MNGIE, potentially driving disease progression. Additionally, we suggest that the gut–liver axis plays a central role in MNGIE pathophysiology, with hepatic function being bidirectionally influenced by gut-derived factors.

## 1. Introduction

Mitochondrial neurogastrointestinal encephalomyopathy (MNGIE) is a progressive and fatal autosomal recessive disorder caused by pathological mutations in the nuclear *TYMP* gene encoding the cytosolic enzyme, thymidine phosphorylase. The consequent total or near total absence of thymidine phosphorylase activity results in systemic accumulations of its substrates, thymidine and deoxyuridine and mitochondrial deoxyribonucleoside triphosphate pool imbalances that lead to mitochondrial DNA (mtDNA) point mutations, deletions, and depletion. Although the exact mechanisms remain unclear, mtDNA abnormalities are believed to disrupt normal mitochondrial function, contributing to the clinical manifestations of MNGIE [1,2,3,4,5].

Clinically, MNGIE is characterised by progressive gastrointestinal dysmotility, cachexia, ptosis, progressive external ophthalmoplegia, sensorimotor peripheral neuropathy and leukoencephalopathy [6,7]. Skeletal muscle biopsies may reveal ragged-red fibres, mtDNA abnormalities, ultra-structurally abnormal mitochondria and defects in mitochondrial respiratory chain enzyme activities, including isolated complex IV deficiency and variable combined deficiencies of complexes I and IV or I, III, and IV [6,8,9]. Multiple case studies involving patients with MNGIE have documented a range of metabolic abnormalities. These include elevated plasma concentrations of pyruvate and lactate, hypertriglyceridemia, and increased urinary excretion of 3-methylglutaconic acid, ethylmalonic acid, creatine, and tricarboxylic acid (TCA) cycle intermediates such as fumarate, aconitate, and 2-oxoglutarate [10,11,12,13,14]. A recent study of Du et al. reported significant alterations in plasma bile acids and steroid metabolites in three patients with MNGIE, two of which were twins [15]. Some of these metabolic alterations indicate a disturbed energy homeostasis, consistent with the impairment of mitochondrial function, and as a result, will affect a broad range of energy-dependent cellular processes, leading to metabolic perturbations across interconnected biochemical pathways. Metabolites are the intermediates or end-products of multiple anabolic and catabolic reactions and therefore represent the most informative proxies of the biochemical activity of an organism. Comparing metabolic profiles between healthy and diseased states allows for the identification of distinct alterations in metabolic pathways and insight into the molecular mechanisms underlying disease pathophysiology. Additionally, dysregulated metabolic pathway expression analyses could reveal information on the drivers of disease heterogeneity and ultimately facilitate the discovery of biomarker panels for supporting the development of effective and personalized treatment approaches.

This study presents the first untargeted metabolomic profiling study of this scale to determine whether patients with MNGIE exhibit a distinct plasma metabolic profile compared to age- and sex-matched healthy controls. Our analysis revealed that the MNGIE patient group exhibited increased activation of the arachidonic acid metabolism pathway and bile synthesis, and identified 23 individual metabolites having potential diagnostic value.

## 2. Results

### 2.1. Study Participants

Each study group recruited 15 participants, matched for sex (10 females and 5 males) and age (16–42 years), with mean ages of 25.7 and 25.5 years for MNGIE patients and healthy controls, respectively.

Patients met the recruitment criteria by harbouring pathogenic mutations in the *TYMP* gene, exhibiting a deficiency of thymidine phosphorylase activity in the buffy coat, and presenting with elevated plasma concentrations of thymidine (>3 µmol/L) and deoxyuridine (>5 µmol/L). All 15 patients demonstrated the characteristic clinical manifestations of MNGIE at the time of recruitment, including peripheral sensorimotor polyneuropathy, external ophthalmoplegia, intestinal dysmotility, and cachexia. To identify differential metabolites between MNGIE patients and healthy controls, plasma samples were analyzed using untargeted ultra-high-performance liquid chromatography–mass spectrometry liquid chromatography (UHPLC-MS) metabolomics in both positive (ESI+) and negative (ESI−) electrospray ionization modes.

### 2.2. Quality Control of UHPLC-MS Analysis

Quality Control (QC) samples were inserted with every ten samples of the analytical batch to evaluate the reproducibility and stability of the analytical system. The quality of metabolite extraction and UHPLC–MS analysis was assessed by base peak chromatogram (BPC) overlay of QC samples, Principal component analysis (PCA) of all samples and coefficient of variations (CVs) of features (retention time and m/z) in QC samples and internal standards.

The BPCs of QC samples in the positive and negative modes demonstrated good overlap, with little fluctuation of retention time and peak response intensity, indicating excellent instrument stability and data quality (Figure 1).

PCA on the quantitative values of all samples in the overall dataset including the QC samples, showed a good separation of the healthy control and disease samples and a high overlap of the QC samples. The separation from the healthy control and disease samples highlights differences due to biological factors, and not technical variability. The latter validated the stability and reproducibility of the UHPLC-MS analysis process (Figure 2a). After peak information extraction and matching, 9920 metabolite characteristic ions were obtained. The repeatability of the QC samples was assessed by calculating the CV for each metabolite’s intensity. The higher the proportion of compounds with low CV value in QC samples is, the more stable the experimental data is. The number of features in QC samples with CVs ≤ 30% were 7774 metabolites, with a percentage of 84% (Figure 2b). These results confirmed that the plasma metabolic characteristics were reproducible and reliable in the analysis workflow.

### 2.3. Metabolite Identification and Classification

Plasma metabolites were annotated with five confidence levels based on information available for identification, including MS1 molecular weight, MS2 fragment spectra, column retention time and whether there are reference standards. The metabolite identification credibility evaluation criteria employed was: level 1: identified results were from standard libraries with the matching of MS1, MS2 spectra and retention time; level 2: structural formula matches the standard database; level 3: structural formula partly matches the standard database, but needs further verification; level 4: identified results were from database libraries, and only MS1 molecular weight can match the theoretical values; and level 5: no match or no identification results for the metabolite in the database.

The 7774 ions that were identified following data preprocessing had the following credibility levels: 60 (level 1), 262 (level 2), 136 (level 3), 1853 (level 4) and 5463 (level 5), resulting in 2311 metabolites with identification information. Of these 2311 metabolites, 1348 were found in the BGI metabolome, Chemspider and mzCloud databases with corresponding compound information. The Kyoto Encyclopedia of Genes and Genomes (KEGG) database and Human Metabolome Database (HMDB) annotated these metabolites into 37 different classes as shown in Figure 3, where lipid metabolite identification was based on their sub class, and the remaining metabolites identified through their final class. Others included venoms (1 entry), organosulfur compounds (11), organoheterocyclic compounds (112), organic oxygen compounds (26), organic nitrogen compounds (7), organohalogen compounds (3), hydrocarbons (5), homogeneous non-metal compounds (6) and fungal toxins (1).

### 2.4. Multivariate Analysis of Identified Metabolites

The PCA model used to perform an unsupervised multivariate analysis of the healthy controls and patient groups of plasma samples shows a clear separation between the healthy control and patient plasma samples, with the combined PCA components 1 and 2 accounting for 41.02% of the variance (Figure 4). All samples in each group were within the 95% confidence interval. To establish a relationship model between sample groups and metabolite expression and realise the modelling and prediction of sample categories, an Orthogonal partial least squares discriminant analysis (OPLS-DA) model was constructed. This analysis showed that the first predictive component T score [1] explained 22.8% variation between the healthy and MNGIE groups (*X*-axis), while the orthogonal T-score [1] accounted for 15.1% of the variation within the groups (*Y*-axis), Figure 4.

The OPLS-DA score plot shows that the healthy control and MNGIE disease groups were clearly separated from each other by the first predictive component T score [1], indicating distinct plasma metabolic profiles due to the disease/healthy state. The results of the OPLS-DA permutation test are shown in Figure 5. Parameters used to evaluate the OPLS-DA model were R^2^Y (cum) which is the interpretation rate for the *Y* matrix; Q^2^ (cum), the predictive capability; and R^2^ and Q^2^ the intercepts of the *Y*-axis for the regression line during permutation experiments. The model produced the following outcomes: R^2^Y (cum) = 0.985 and Q^2^ (cum) = 0.919, indicating the model is stable and reliable. Q^2^ was higher than 0.5 indicating that the model’s predictive performance was good. The permutated values of R^2^ and Q^2^ on the *Y*-axis were (0.0, 0.84) and (0.0, −0.34), respectively, and were significantly lower than the original R^2^ and Q^2^ values on the right demonstrating that the model was valid and not over-fitted. The overall contribution of each variable was ranked in the OPLS-DA model. Those metabolites with a variable influence on projection (VIP)  ≥  1 were selected for group discrimination.

### 2.5. Differential Metabolite Identification

To identify plasma metabolites that have the highest discriminative power between the healthy and control groups, the multivariate analysis model for VIP was combined with univariate analyses. Using the criteria of fold-change  ≥ 1.2 or ≤0.83 (selected on the basis that when log_2_ transformed, 1.2 and ≤0.83 are equidistant from zero, the point of no change), *q*-value  ≤  0.05, and VIP  ≥  1 for differential metabolites, a total of 317 metabolites were identified as upregulated and 397 identified as down-regulated in patients with MNGIE, compared to healthy controls. Figure 6 presents a volcano plot of this analysis, where the abscissa represents the logarithm of the fold change in the differential metabolites [log_2_ (Fold Change)], and the ordinate represented the negative logarithm of *q*-value [−Log 10 (*q*-value)].

The metabolite datasets were further filtered to remove exogenous compounds, including drugs, drug metabolites, and environmental exposures. In total, 160 upregulated metabolites remained, consisting of 7, 39, 12 and 102 metabolites, respectively, at credibility levels 1, 2, 3 and 4, (Appendix A). Most of the up-regulated metabolites identified at credibility levels 1 and 2, were classified as lipids and lipid-like molecules (35 metabolites), with the remaining classified as nucleosides, nucleotides, and analogues (2 metabolites), organic acids and derivatives (5 metabolites), organic oxygen compounds (1 metabolite) and organoheterocyclic compounds (3 metabolites), Appendix A.

For the downregulated dataset, 219 metabolites remained after exogenous metabolite exclusion, consisting of 4, 25, 6 and 184 metabolites at credibility levels 1, 2, 3 and 4, respectively (Appendix A). Down-regulated metabolites at credibility levels 1 and 2 were classified as follows: Benzenoids (1 metabolite), Fatty acyls (1 metabolite), lipids and lipid-like molecules (12 metabolites), organic acids and derivatives (6 metabolites), organoheterocyclic compounds (8 metabolites), and phenylpropanoids and polyketides (1 metabolite), Appendix A.

The abundances of the differential metabolites were normalized using z-scores and then subjected to hierarchical clustering analysis. The results from this analysis showed significant differences between healthy control and MNGIE plasma samples, with samples from the same group clustering, Figure 7.

### 2.6. Metabolic Pathway Enrichment Analysis of Differential Metabolites

To contextualise the biological phenotype of MNGIE, a KEGG pathway enrichment analysis was performed on the filtered differential metabolites. Based on a *p* value < 0.05 there was an enrichment of the annotated metabolites in 20 pathways (Figure 8). The top pathways with the highest enrichment were caffeine metabolism; steroid hormone biosynthesis; pathways in cancer; arachidonic acid metabolism; ovarian steroidogenesis; pyrimidine metabolism; prostate cancer; alanine, aspartate and glutamate metabolism; and serotonergic synapse. Of the 379 differential metabolites that were in the final dataset, 54 could be assigned to one or more KEGG pathways, as shown in Table 1.

Insight into the broader perturbations in the metabolite levels within the 20 pathways, was obtained by conducting a differential abundance score analysis, where the score captures the overall changes in all differential metabolites in a pathway [16]. A score of 1 indicates that all measured metabolites in a pathway are upregulated, and −1 indicates all measured metabolites in a pathway are down regulated. As shown in Table 1, 9 pathways were downregulated with the lowest differential abundance scores associated with pathways involving caffeine metabolites and steroid hormones. Eleven pathways were upregulated, with the highest abundance scores associated with pathways involving arachidonic acid, pyrimidine and unsaturated fatty acid metabolites.

### 2.7. Discriminative Ability of Differential Plasma Metabolites for MNGIE Disease

To assess the discriminative power of the differential metabolites, ROC curves were generated and the AUC values calculated for metabolites identified based on a *q*-value < 0.05, fold change ≥ 1.2, VIP ≥ 1, and confidence levels between 1 and 2. Figure 9 shows the discriminatory power of potential biomarkers citrate, itaconate, 5,6-dihydrourcail, deoxycholate, cholate, retinoate and creatine, all having level 1 credibility evidence and thus confirmed with authentic standards, and AUCs ranging between 0.80 and 0.95. The boxes of the box and whisker plots denote interquartile ranges, with the horizontal line inside the box showing the median, and bottom and top boundaries of boxes representing 25th and 75th percentiles, respectively. Lower and upper whiskers are 5th and 95th percentiles, respectively, and single datapoint outliers are shown as circles.

Ten lipid metabolites at credibility level 2 could be discriminated between the healthy and control groups, with AUC values ranging between 0.80 and 0.97 (Figure 10).

Other potential biomarkers upregulated in MNGIE at credibility level 2 are shown in Figure 11, with AUC values ranging between 0.80 and 0.97. As expected, the performances of thymidine and deoxyuridine were confirmed, both established biomarkers of MNGIE, showing AUC values of 0.99 and 0.97, respectively.

The involvement of these biomarkers in various metabolic pathways is depicted in Figure 12.

## 3. Discussion

In this study we conducted an untargeted metabolomics analysis of intravenous plasma samples to identify metabolite changes in patients with MNGIE, compared to age- and sex-matched healthy controls.

The elevated levels of creatine, lactate, citrate, itaconate, trans-aconitate, and fumarate observed in this study suggest a disrupted intracellular energy state, consistent with mitochondrial dysfunction. Notably, increased plasma creatine levels have been previously reported in several mitochondrial disorders, including Mitochondrial Encephalomyopathy, Lactic Acidosis, and Stroke-like Episodes syndrome, Myoclonic Epilepsy with Ragged Red Fibers syndrome, Leber’s Hereditary Optic Neuropathy (LHON), and various mitochondrial DNA (mtDNA) depletion and deletion syndromes [17,18,19,20]. Creatine plays a central role in maintaining ATP concentrations in tissues with high energy demands such as skeletal muscle, cardiac muscle and brain through its conversion to phosphocreatine by intracellular creatine phosphokinase. Patients with respiratory chain disorders often exhibit reduced intramuscular concentrations of phosphocreatine and adenosine triphosphatase activity. Oral creatine supplementation is frequently prescribed to individuals with respiratory chain disorders as a strategy to enhance muscle energy metabolism, although the therapeutic benefits remain inconclusive [21,22,23,24,25,26]. For patients with MNGIE, baseline plasma creatine measurement may be informative for predicting whether creatine supplementation is likely to be effective. While patients in the current study were not prescribed creatine supplements, we cannot exclude the possibility that elevated plasma creatine levels may reflect self-administration of dietary supplements. Additionally, our patient cohort exhibited elevated levels of 4-guanidinobutyric acid, a metabolite of arginine and an intermediate in creatine biosynthesis. Importantly, 4-guanidinobutyric acid has been identified as an alternative substrate for the creatine transporter-1 (CRT1/SLC6A8), and at high concentrations, it can inhibit cellular creatine uptake [27]. This raises the question: could this mechanism contribute to the elevated plasma creatine levels observed in MNGIE? If so, what might be the potential consequences of an intracellular accumulation of 4-guanidinobutyric acid?

The elevated levels of lactate in our patient group agree with previous findings in patients with MNGIE and are accordant with an enhanced glycolytic flux, a common feature of mitochondrial disorders where oxidative phosphorylation is compromised [11,13,15,28,29]. The increases in citrate, fumarate, itaconate and trans-aconitate levels reported here indicate disturbances in the tricarboxylic acid cycle. Individual case studies have also reported elevated TCA cycle-related metabolites in MNGIE, including isocitrate, aconitate and fumarate [13,15]. Elevated blood levels of citrate are associated with liver fibrosis in conditions such as metabolic dysfunction-associated steatotic liver disease (MASLD) and metabolic dysfunction-associated steatohepatitis [30]. Hepatopathy presenting as steatosis, hepatomegaly and cirrhosis are features of MNGIE and as citrate is normally retained within the mitochondria for ATP production, the increase in plasma levels reported in our study may be due to an enhanced release from damaged hepatocytes and/or a reduced uptake by the liver [6,28]. Other factors worthy of consideration are the gut microbiota which has been shown to be a source of circulating TCA cycle intermediates, and patients with cirrhosis in general often have imbalances in their microbiome [31,32,33]. Patients with MNGIE frequently exhibit intestinal bacterial overgrowth, leading to microbiome imbalances, due to the gastrointestinal dysmotility aspect of the disease [8]. The elevated circulating TCA cycle intermediates observed in MNGIE may therefore be linked to gut microbiota dysregulation.

Itaconate is derived from the decarboxylation of the tricarboxylic acid cycle intermediate, cis-aconitate, and has been widely shown to exert anti-inflammatory responses by regulating macrophage immune and metabolic activities through multiple pathways [34,35,36]. An example of this is demonstrated in the study of Weiss et al. who reported the immunomodulating effect of up-regulated levels of macrophage derived itaconate in human MASLD and a mouse model of non-alcoholic fatty liver disease. Macrophage-derived itaconate was shown to act upon hepatocytes to modulate the liver’s ability to metabolize fatty acids [37]. In light of the hepatopathological aspects of MNGIE, it could be hypothesized that the elevated levels of itaconate observed in our study represent a compensatory mechanism to promote hepatocyte lipid oxidation.

Our study showed a significant dysregulation in lipid metabolism, with the upregulation of two polyunsaturated fatty acids (docosahexaenoic acid and 8,11,14-eicosatrienoic acid), 16 arachidonic acid derivatives, two docosahexaenoic acid derivatives, two 8,11,14-eicosatrienoic acid derivatives, three eicosapentaenoic acid derivatives, and three linoleic acid derivatives. An enrichment analysis revealed that arachidonic acid metabolism was the most significantly up-regulated pathway in our patient group, with elevated levels of the eicosanoids lipoxin B4, series 2 prostaglandins, series 2 thromboxanes, and derivatives of hydroperoxyl, hydroxy and epoxy eicosatrienoic acid. Eicosanoid biosynthesis in general is initiated by an activation signal, leading to the release of polyunsaturated fatty acids from membrane glycerophospholipids by cytoplasmic phospholipases. The released polyunsaturated fatty acids are further metabolised to various eicosanoids by several enzymes including cyclooxygenases, P450 cytochrome epoxygenases and lipoxygenases [38]. It is well established that eicosanoids are highly bioactive signalling molecules that play critical roles in regulating physiological processes including thrombosis, vascular tone and the initiation and resolution of acute inflammation. However, imbalances between the production of pro-inflammatory and pro-resolving eicosanoids have been shown to underpin chronic inflammation, leading to tissue dysfunction and the development of disease [39]. As well as an up-regulation of proinflammatory eicosanoids, our patient cohort revealed elevated levels of pro-resolving eicosanoids (also known as Specialized Pro-resolving Mediators) including the eicosapentaenoic acid -derived resolvin E1, docosahexaenoic acid- derived protectin D1, and as already mentioned above, arachidonic acid-derived lipoxin B4. This may indicate that pro-resolving mechanisms were activated in response to an inflammatory state in our patient cohort but were either not sufficiently potent to resolve the inflammation or there was an impaired responsiveness to these pro-resolving signals. We propose that the mechanisms responsible for resolving inflammation are impaired in MNGIE, resulting in a persistent inflammatory state and contributing to disease progression over time. If this hypothesis holds true, it may have important implications for the use of parenteral nutrition, which is commonly administered to address the malnutritional aspects of MNGIE [40,41,42,43]. Notably, linoleic acid, a major component of lipid-based parenteral emulsion, is metabolized into arachidonic acid, a precursor of proinflammatory mediators, and could therefore potentially exacerbate inflammatory responses. In contrast, dietary supplementation with docosahexaenoic acid and eicosapentaenoic acid has been shown to elevate levels of pro-resolving eicosanoids in the peripheral blood of healthy individuals, while also reducing hepatic steatosis and improving lipid profiles in patients with MASLD [44]. For patients with MNGIE who are at risk of metabolic oversupply due to components of parenteral nutrition, and who may have impaired biosynthesis of pro-resolving eicosanoids, the therapeutic administration of these mediators could offer significant benefits by mitigating chronic inflammation and providing hepatoprotective effects.

An intriguing observation in our study is that patients exhibited significantly elevated levels of 2-arachidonoylglycerol compared to healthy controls. As an endocannabinoid, 2-arachidonoylglycerol activates CB1 receptors and is primarily metabolized into arachidonic acid and glycerol. Emerging evidence indicates that 2-arachidonoylglycerol plays a central role in regulating hepatic triglyceride synthesis and release. Elevated 2-arachidonoylglycerol concentrations and subsequent CB1 receptor hyperactivation have been linked to impaired glucose metabolism in skeletal muscle and increased hepatic lipid accumulation, both of which are key contributors to the development of insulin resistance and non-alcoholic fatty liver disease, and are pathologies associated with MNGIE [28,45,46,47,48].

Pyrimidine metabolism was identified as the second most upregulated metabolic pathway in our patient cohort. As expected, the two hallmark biomarkers of MNGIE, thymidine and 2′-deoxyuridine, were markedly elevated. The products of the forward reaction catalyzed by thymidine phosphorylase, thymine and uracil, were also increased, consistent with previous reports by Vondráčková et al. and more recently by Du et al. [13,15]. Under normal conditions, thymine and uracil are metabolized by dihydropyrimidine dehydrogenase to 5,6-dihydrothymine and 5,6-dihydrouracil, respectively. Thymidine has been reported as a non-competitive inhibitor of dihydropyrimidine dehydrogenase, which could account for the accumulation of thymine and uracil. However, this mechanism does not readily explain the elevated levels of 5,6-dihydrouracil observed in our patient cohort [49]. Notably, previous studies have reported higher plasma concentrations of 5,6-dihydrouracil in the fasting state compared to the fed state which are believed to be the direct result of the homeostatic control of uridine [50]. Thus, differences in nutritional status between patients with MNGIE and healthy controls may explain this discrepancy.

Another notably upregulated pathway identified in our study was bile acid biosynthesis, characterized by elevated levels of unconjugated bile acids, cholate, β-muricholic acid and deoxycholate. While cholate and β-muricholic acid are primary bile acids synthesized in the liver from cholesterol and secreted into the duodenum, deoxycholate is a secondary bile acid formed through microbial enzymatic conversion of cholate in the intestinal tract [51]. β-muricholic acid was elevated by 19-fold in our patient cohort compared to healthy controls. This finding is particularly intriguing, as β-muricholic acid, although abundant in rodents, is typically detected at only low concentrations in humans [52]. Nevertheless, a study by Cui et al. reported a significant increase in serum β-muricholic acid levels in women with intrahepatic cholestasis of pregnancy, compared to healthy pregnant controls [53]. Additionally, other studies have shown that β-muricholic acid is not metabolised by human intestinal bacteria [54]. Therefore, a combination of dysregulated bile acid biosynthesis and the inability to metabolise β-muricholic acid is likely to explain the detected elevated levels observed in our patient cohort. Consistent with our findings, Du et al. recently reported elevated levels of cholate in three patients with MNGIE, along with three other bile acids, 3b,4b,7a,12a-tetrahydroxy-5b-cholanoic acid, hyocholic acid and chenodeoxycholic acid [15]. Although dysregulated bile acid homeostasis is closely associated with liver disease, the cause-and-effect relation between the hepatopathy aspects of MNGIE and the observed dysregulated bile acid synthesis remains unclear. While liver dysfunction is known to impair bile acid metabolism, elevated circulating bile acids can themselves exert hepatotoxic effects. This toxicity arises from their surfactant-like action on membrane phospholipids, which can trigger phospholipase A2-mediated inflammation and the release of pro-inflammatory mediators [55]. The upregulation of pro-inflammatory mediators in our patient cohort, as discussed above, may therefore reflect bile acid-induced disruption of cellular membrane integrity. Additionally, because normally the liver excretes bile acids only in the conjugated form, unconjugated bile acids in the peripheral circulation are attributable either to bacterial metabolic action or as a result of cholestasis leading to bile acid escape into the bloodstream. Unconjugated bile acids have been reported to be elevated in the serum of patients exhibiting clinical signs of small intestinal bacterial overgrowth, a known complication of MNGIE due to intestinal dysmotility [33,56]. Moreover, disruptions in bile acid metabolism may contribute to gastrointestinal complications in MNGIE by triggering inflammation of the intestinal epithelium, potentially leading to gastrointestinal perforations [28]. The study by Smirnova and colleagues demonstrated that serum levels of deoxycholic acid increased progressively with the severity of MASLD, suggesting its potential as a biomarker for liver disease progression [57]. Thus, the gut-liver axis may play a key role in MNGIE pathology, with hepatic function being bidirectionally influenced by gut activity. Our ROC analyses highlighted the strong discriminatory power of cholate, β-muricholic acid, and deoxycholate in distinguishing patients from healthy controls, with AUC values ranging from 0.80 to 0.83. These findings suggest that the identified metabolites may serve as promising metabolic biomarkers for monitoring the hepatopathological features of MNGIE. A biomarker panel incorporating these metabolites may help optimise the timing of liver transplantation by identifying biochemical indicators before the onset of advanced liver disease. This is particularly relevant given that, at the time of recruitment, our patient cohort did not exhibit signs of liver dysfunction; however, four individuals subsequently developed severe hepatic abnormalities.

The second most significantly downregulated pathway identified in our analysis was steroid hormone biosynthesis. Notably, several metabolites involved in this pathway also participate in other signaling cascades, including Pathways in cancer, Ovarian steroidogenesis, and Prolactin signaling, which were likewise downregulated. The down-regulation of pathways in cancer is noteworthy, particularly given that thymidine phosphorylase is upregulated across a broad spectrum of solid tumours, where its elevated expression is associated with increased tumour growth, metastasis, angiogenesis, and poor prognosis. Whether the absence of thymidine phosphorylase expression confers protection against cancer development remains to be determined. Notably, we are not aware of any reported cases of cancer in patients with MNGIE. The study by Du et al. also revealed reduced plasma levels of several steroids, including 4-methoxyestrone, 17α-hydroxypregnenolone, androsterone sulphate, and dehydroepiandrosterone sulphate [15]. However, in contrast to our findings, they reported elevated levels of androstenedione. Given that perturbations in mitochondrial membrane potential and ATP synthesis have been shown to inhibit steroidogenesis, it is plausible that, in the context of MNGIE, cholesterol is preferentially redirected toward bile acid synthesis [58]. This metabolic shift may reduce the availability of cholesterol for steroid hormone production, thereby contributing to the observed suppression of steroid biosynthesis and its associated pathways.

ROC curve analysis of the differential metabolites identified 23 candidate biomarkers with high diagnostic potential for MNGIE. Among these findings, the 11-fold increase in retinoate abundance is particularly striking. Retinoate has been widely reported to inhibit adipogenesis [59,60,61]. In our recent study, we demonstrated that *TYMP* knockout in adipose stem cells disrupts adipocyte differentiation. We proposed that the marked thinness observed in patients with MNGIE is not solely a consequence of cachexia secondary to gastrointestinal dysfunction, but also reflects lipoatrophy driven by a primary defect in adipose tissue [62]. This observation highlights a potential contribution of retinoate to MNGIE pathophysiology and underscores the need for further investigation into its mechanistic role. Furthermore, given that lipids were the most dysregulated metabolites in this study and that the extraction method employed has some limitations in the recovery of certain lipid species, a dedicated lipidomic analysis is warranted. Moreover, verification and validation studies involving larger independent cohorts are essential to confirm the diagnostic utility of these 23 metabolites and to further elucidate underlying pathophysiological mechanisms in MNGIE.

Several limitations of our study should be acknowledged.

Influence of the exposome: Metabolic profiles are significantly shaped by the exposome, which includes factors such as diet, dietary supplements, medicinal and recreational drugs, personal care products, and occupational exposures. Although known exposome-related metabolites were excluded from our dataset, we cannot entirely rule out the possibility that these exposures influenced the observed metabolic profiles. MNGIE patients typically receive a combination of medications, including analgesics, bowel motility stimulants, anti-emetics, antibiotics, and centrally acting agents tailored to their symptoms. In contrast, healthy controls may be exposed to different confounding factors. Notably, caffeine metabolism emerged as the most significantly downregulated pathway in MNGIE patients. Caffeine is mainly absorbed by the small intestine and is metabolized via demethylation and/or hydroxylation into paraxanthine, theobromine, theophylline, and 1,3,7-trimethyluric acid, all of which were significantly reduced in our patient cohort. Patients with MNGIE exhibit severe gastrointestinal manifestations, including dysmotility, abdominal pain, nausea, dysphagia, pseudo-obstruction, and diarrhoea. These symptoms frequently lead to oral intolerance, progressive weight loss, and malnutrition. The observed differences in caffeine metabolism may reflect comparatively higher caffeine intake among healthy controls.Sample size constraints: MNGIE is an ultra-rare disorder with an estimated prevalence of fewer than 1 in 1,000,000 individuals in Europe. Since its initial description by Okamura et al. in 1979, approximately 500 cases have been reported globally [63]. The mean life expectancy is 35 years. Thus, the rarity of MNGIE presents a significant challenge for clinical research, particularly in achieving adequate sample sizes for statistically robust analyses.

In this study, we investigated the metabolic phenotype of MNGIE using a cohort of 15 patients and 15 matched controls. While the limited sample size is a recognized constraint, we implemented several strategies to maximize statistical power and ensure data reliability:Careful recruitment of age- and sex-matched controls to minimize confounding demographic effects on the metabolome.Strict adherence to standardized protocols for sample collection, processing, and storage to reduce pre-analytical variability and ensure that observed metabolic alterations reflect disease-specific changes.Application of high-resolution mass spectrometry to enhance analytical sensitivity and selectivity.Use of pooled QC samples to monitor instrument performance and correct for technical variability.Comprehensive statistical analysis, including PCA and OPLS-DA, to reduce data dimensionality and identify meaningful metabolic patterns.Implementation of the Benjamini–Hochberg procedure to control the false discovery rate (FDR) and reduce the likelihood of type I errors.

Due to the cross-sectional nature of the study and the limited number of patients, we were unable to assess the influence of disease progression on the metabolic profile. Future longitudinal studies are essential to characterize metabolomic dynamics over time and in response to therapeutic interventions.

3.Lack of targeted metabolite validation: The findings from our untargeted metabolomics analysis were not complemented by targeted metabolite quantification. Such validation will be essential to confirm the diagnostic relevance of the 23 identified biomarkers. However, the computational integration of biomarkers across multiple omics layers, such as the genome, transcriptome, proteome, and metabolome, is more likely to yield insights into underlying molecular and cellular mechanisms and may identify more effective measures of treatment outcomes than relying on a single biomarker. Indeed, integrative analysis approaches can be employed to correlate multi-omics data with disease phenotypes, treatment responses, and patient stratification.4.Analytical scope limitations: Since no single chromatography column can separate all metabolites in a sample, we selected a C18 column for this study, as it offers relatively broad coverage for separating non-polar and moderately polar metabolites. Consequently, metabolic disturbances involving highly polar or strongly hydrophilic compounds would not have been captured.5.Annotation challenges: Chemical entities assigned to credibility levels 3 to 5 could not be further analyzed due to current limitations in metabolite classification and annotation databases.

## 4. Materials and Methods

### 4.1. Study Participants

A total of 15 patients with MNGIE and 15 age- and sex-matched healthy controls were recruited into this study. The inclusion criteria for patient eligibility were a definitive diagnosis of MNGIE due to thymidine phosphorylase deficiency based upon DNA sequencing, and/or <10% of normal thymidine phosphorylase activity in the buffy coat, and biochemical criteria. Exclusion criteria for both patients and healthy controls included participation in a controlled trial of an investigational medicinal product, receipt of blood transfusions within the past 4 months, feeding via total parenteral nutrition and a current or history of hepatitis B, hepatitis C, or human immunodeficiency virus infection. The study was approved by the NHS Research Ethics Committee (London–Surrey Borders, reference 18/LO/2173) and was conducted in accordance with the Declaration of Helsinki and Good Clinical Practice Guidelines. Written informed consent was obtained from all participants prior to blood collection.

### 4.2. Blood Collection

Ten mL of venous blood were collected from each participant into a K2EDTA-treated BD Vacutainer tube (Beckman Dickinson, Franklin Lakes, NJ, USA) using a standard phlebotomy protocol. Blood was collected in the morning, 4–5 h postprandial and processed immediately after collection using a standardized study protocol. Briefly, blood samples were mixed by gently inverting the tubes 180° and back, 10 times and then centrifuged at 1500× *g* for 10 min at 4 °C. The plasma supernatants were aspirated using sterile pipette tips and apportioned into sterile cryotubes as 0.5 mL aliquots and then stored at −80 °C until required for metabolite extraction. Samples which showed signs of haemolysis were excluded from the study. Plasma was selected over serum for this metabolomics study to avoid variability from platelet metabolism during serum collection, which is especially relevant given the fluctuating platelet counts in MNGIE.

### 4.3. Untargeted Metabolomic Profiling

Metabolite profiling was conducted according to the workflow, presented in Figure 13 and described in detail below.

#### 4.3.1. Metabolite Extraction

Frozen plasma samples were thawed at 4 °C. Metabolites were extracted from 100 μL of each plasma sample by adding 700 μL of pre-chilled (−20 °C) quenching/extraction solvent (methanol: acetonitrile: water in a ratio of 4:2:1, *v*/*v*/*v*) containing 10 µL of internal standard mix (0.3 mg/mL d3-leucine, 13C9-phenylalanine, d5-tryptophan, and 13C3-progesterone) for QC of sample preparation. Samples were vortexed for 1 min, incubated at −20 °C for 2 h and then centrifuged at 25,000× *g* for 15 min. Six hundred µL of each supernatant were transferred to a new tube for vacuum freeze drying. The dried metabolite extracts were reconstituted in 180 μL of 50% methanol, vortexed for 10 min and then centrifuged at 25,000× *g* for 15 min. The reconstituted sample extracts were subsequently transferred to autosampler vials for UHPLC-MS analysis. Eleven QC samples (QC samples 6–16) were prepared by pooling 20 µL of each plasma sample. Aliquots of the pooled samples were then processed in an identical approach as the plasma test samples. A set of five QC samples were injected at the beginning of the run to condition the system for the study matrix. The samples were randomly ordered to decrease system errors and for every nine to ten samples, a QC sample was interposed, ending with three QC samples.

#### 4.3.2. UHPLC-MS Analysis

Extracted metabolites were separated and detected using an UHPLC system (Waters I-Class Plus, Milford, MA, USA), interfaced with a Q Exactive high-resolution mass spectrometer (Thermo Fisher Scientific, Waltham, MA, USA).

Chromatographic separation was performed on a Waters ACQUITY UPLC BEH C18 column (1.7 μm, 2.1 mm × 100 mm, Waters, Milford, MA, USA) maintained at a temperature of 45 °C. The mobile phase consisted of 0.1% formic acid (A) and acetonitrile (B) in positive mode, and 10 mM ammonium formate (A) and acetonitrile (B) in the negative mode. The gradient conditions were: 0–1 min, 2% B; 1–9 min, 2–98% B; 9–12 min, 98% B; 12–12.1 min, 98% B to 2% B; and 12.1–15 min, 2% B. The flow rate was 0.35 mL/min and the injection volume was 5 µL.

The mass spectrometer (MS) was adjusted for positive/negative electrospray ionization modes as follows: 3.8/−3.2 kV spray voltage; 40 arbitrary units (arb) sheath gas flow rate; 10 arb aux gas flow rate; 350 °C aux gas heater temperature; and 320 °C capillary temperature. For MS acquisitions, the scan range was 70–1050 *m*/*z* with a resolution of 70,000 and the automatic gain control (AGC) target was set to 3 × 10^6^ with a maximum ion injection time of 100 ms. The top three precursors were selected for subsequent MS/MS fragmentation with a maximum ion injection time of 50 ms, resolution of 17,500, and AGC of 1 × 10^5^.

#### 4.3.3. Metabolite Ion Peak Extraction and Metabolite Identification

The raw data collected from the mass spectrometry analysis were imported into Compound Discoverer 3.2 (Thermo Fisher Scientific, USA, https://mycompounddiscoverer.com/) software for data processing, accessed 12 November 2024. This included peak extraction, retention time (RT) correction, additive ion pooling, missing value filling, background peak labeling and metabolite identification. Metabolite identification was conducted in combination with the BGI metabolome database (bmdb, Shenzhen, China), mzcloud (Thermo Fisher Scientific) and ChemSpider online database. Compound Discoverer 3.2 software parameters for metabolite identification were set as follow: parent ion mass tolerance < 5 ppm, fragment mass tolerance < 10 ppm, and retention time tolerance < 0.2 min.

Metabolites were categorized into five confidence levels based on information available for identification, including MS1 molecular weight, MS2 fragment spectra, column retention time and whether there are reference standards. The metabolite identification credibility evaluation criteria employed was: Level 1: identified results were from standard libraries with the matching of MS1, MS2 spectra and retention time; Level 2: structural formula matches the standard database; Level 3: structural formula partly matches the standard database, but needs further verification; Level 4: identified results were from database libraries, and only MS1 molecular weight can match the theoretical values; and Level 5: no match or no identification results for the metabolite in the database.

#### 4.3.4. Data Preprocessing

Files generated by Compound Discoverer 3.2 containing information on molecular weight, retention time, peak area, and identification were exported to metaX (http://metax.genomics.cn, accessed 12 November 2024), the metabolomics R package, for data preprocessing and statistical analyses [64]. During pre-processing, the data were normalized to obtain relative peak areas by Probabilistic Quotient Normalization, PQN [65]. Then, batch effects were corrected using QC-based robust LOESS signal correction (QC-RLSC) [66]. Metabolites with CV larger than or equal to 30% on their relative peak area in QC samples were removed from further analysis.

#### 4.3.5. Data QC

QC samples were injected at regular intervals throughout the analytical run to monitor the retention time (RT/min) shift and signal variations and thus evaluate the reproducibility of the LC-MS analysis process. The quality of metabolite extraction and UHPLC–MS analysis was assessed by BPC overlay of QC samples, PCA of all samples and CVs of features (retention time and *m*/*z*) in QC samples and internal standards.

#### 4.3.6. Classification and Functional Annotation of Detected Metabolites

Identified metabolites were annotated by querying the HMDB and KEGG, followed by functional annotation of pathways using the KEGG pathway database to identify the primary biochemical metabolic pathways and signaling transduction pathways in which metabolites are implicated.

#### 4.3.7. Statistical Analyses

Multivariate statistical and univariate analyses were used to screen the differential metabolites between the healthy control and MNGIE patient sample groups. Prior to multivariate statistical analysis the pre-processed data were log_2_-transformed and auto-scaled in the Pareto scale. An unsupervised PCA was conducted to reduce the dimensionality of original multivariate data and to evaluate separation trends between the two sample groups, intragroup aggregation and to verify clustering of QC samples. A high overlap of QC data is indicative of a reproducible LC-MS analysis process.

OPLS-DA, a supervised statistical method, was performed on the two groups of samples to establish a relationship model between metabolite expression and sample groups, thereby allowing for prediction of the sample categories. To evaluate the robustness of the detected OPLS-DA model, R^2^ Y (cum) and Q^2^ (cum) were calculated using a 7-fold cross-validation, which independently represents the explanatory power and the predictive power of the model. Values of R^2^Y (cum) and Q^2^ (cum) close to 1, would indicate that the model is stable and reliable. To assess the quality of the model without overfitting, a 200-response permutation testing (RPT) was performed. The ability of each metabolite to classify and distinguish each group of samples was measured by calculating the score value of VIP. For the screening of metabolic biomarkers, it is generally considered that a VIP greater than 1 indicates that the metabolite significantly affects the differentiation of the sample groups.

For univariate analysis of the data, the differences in metabolite concentration were evaluated in terms of fold change, and statistical comparison using the *t*-test. Fold change was obtained by fold change analysis, the *p*-value was obtained by Student’s *t*-test, and *q*-value by the Benjamini–Hochberg correction of the *p*-value. The *q*-value was used to assess whether there was a statistical significance between the two analysis groups. The differential metabolites were identified according to the following criteria: VIP of OPLS-DA model ≥ 1, fold change ≥ 1.2 or ≤0.83, and *q*-value < 0.05.

#### 4.3.8. Bioinformatics Analysis

For differential metabolites, with log_2_ transformation and z-score normalized, hierarchical clustering was used as the clustering algorithm and the distance calculation was performed in Euclidean distance. Metabolic pathway enrichment analyses of differential metabolites were conducted by querying the KEGG database (https://www.genome.jp/kegg/pathway.html, accessed 20 May 2025). Significantly enriched pathways were identified with a *p* value < 0.05. ROC analysis was conducted to evaluate the discriminatory power of differential metabolites, with AUC values indicating the following levels of discriminatory power: below 0.7 denoting limited power; 0.7 to 0.8 moderate efficacy, 0.8 to 0.9 good identification accuracy, and 0.9 to 1.0 excellent predictive capability ref below. Putative biomarkers were identified based on *q*-value  <  0.05, fold change  ≥  1.2, VIP  ≥  1, AUC 0.80–1.0, and confidence Levels 1–2.

## 5. Conclusions

In conclusion, this study reveals that the metabolomic profile of MNGIE is characterized by elevated levels of bile acids and arachidonic acid metabolites, alongside reduced levels of steroids, offering insight into disrupted metabolic pathways. ROC curve analysis of the differential metabolites identified 23 candidate biomarkers with high diagnostic potential for MNGIE. These findings suggest that thymidine phosphorylase deficiency leads to broader metabolic disturbances than previously recognized and supports the existence of an expanded biomarker profile for MNGIE.

## Figures and Tables

**Figure 1 ijms-26-09107-f001:**
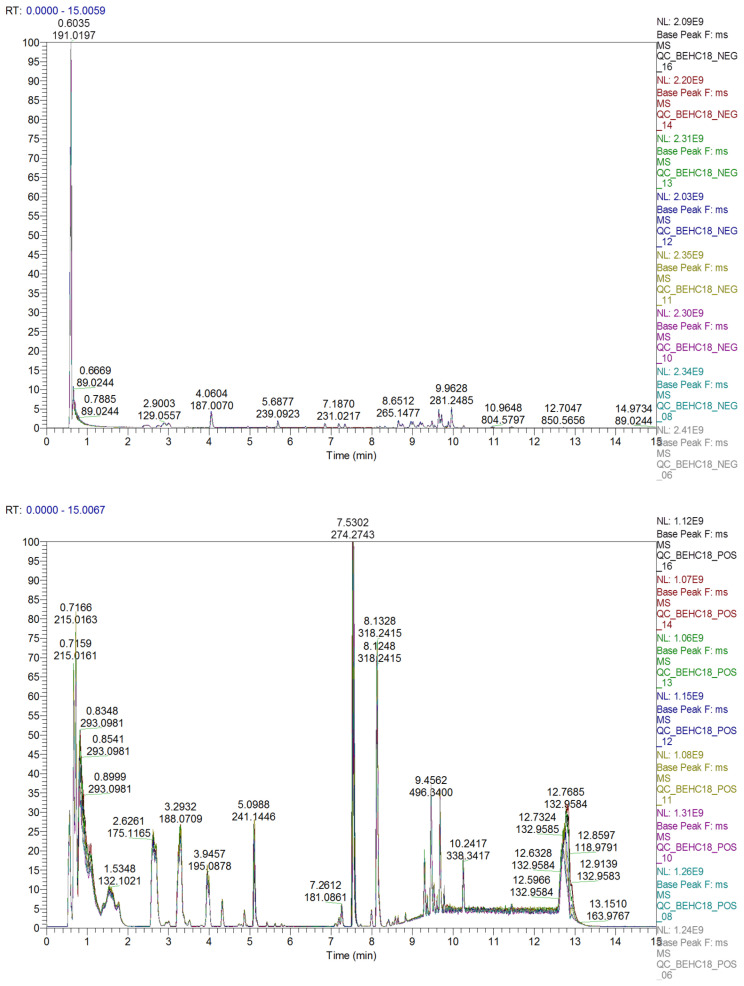
Overlap of the base peak chromatograms of quality control samples in negative (**top figure**) and positive (**bottom figure**) ion modes. The horizontal axis represents the retention time (in minutes) used for metabolite detection, whereas the vertical axis represents the ion response intensity.

**Figure 2 ijms-26-09107-f002:**
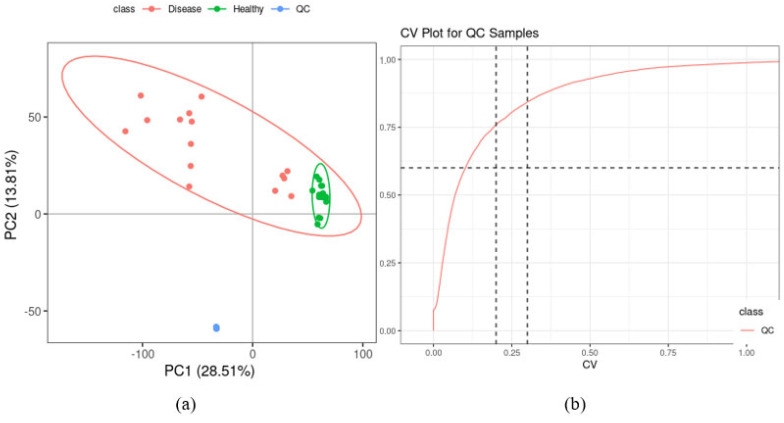
Data quality control. (**a**) PCA score plots for all samples, including the quality control (QC) samples. The scores plot shows separate clusters for disease, control and QC samples thus validating the stability and reproducibility of the UHPLC-MS analysis process. (**b**) CV distribution of compounds in each QC sample. The two lines perpendicular to the *X*-axis represent 20% and 30% CV guides, while the line parallel to the *X*-axis represents the 60% CV guide. The proportion of compounds with a CV value less than 0.3 in QC samples was 84%, therefore qualifying the data quality.

**Figure 3 ijms-26-09107-f003:**
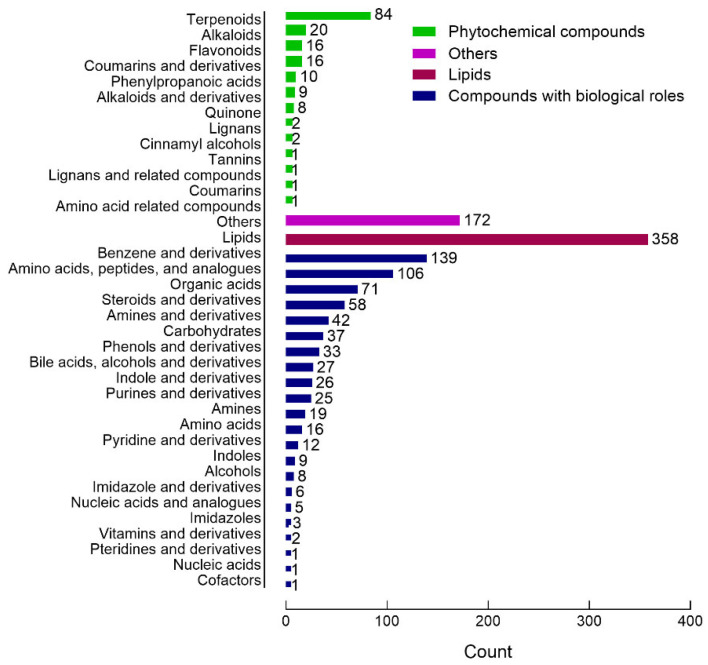
Bar chart of metabolites identified in combined patient and healthy control samples, annotated using the HMDB and KEGG database. Lipid metabolite identification was based on their sub class, and the remaining metabolites identified through their final class. The *X*-axis represents the number of metabolites in each class, and the *Y*-axis represents the metabolite classification entries. The results show that 1348 molecular features were classified into four categories, including compounds with biological roles (n = 647), lipids (n = 358), phytochemical compounds (n = 171), and others (n = 172).

**Figure 4 ijms-26-09107-f004:**
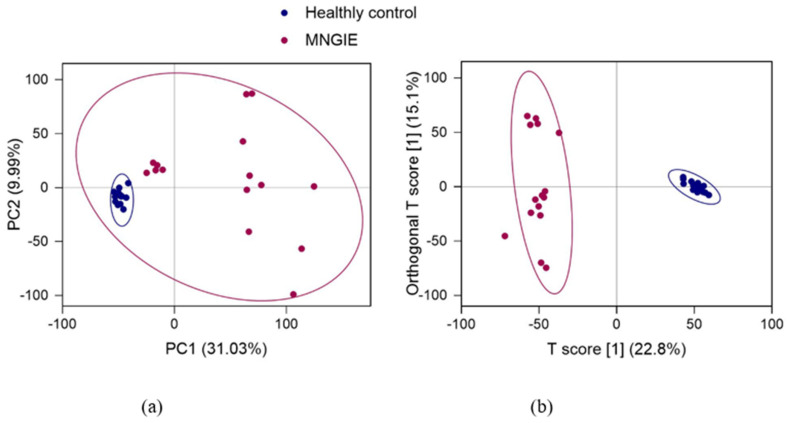
PCA and OPLS-DA score plots between healthy control and MNGIE disease groups. Data were log- transformed (log_2_) and scaled using the Pareto scaling method prior to analysis. (**a**) PCA model and (**b**) OPLS-DA analysis model. Each point represents a sample, and the ellipses represent 95% confidence intervals.

**Figure 5 ijms-26-09107-f005:**
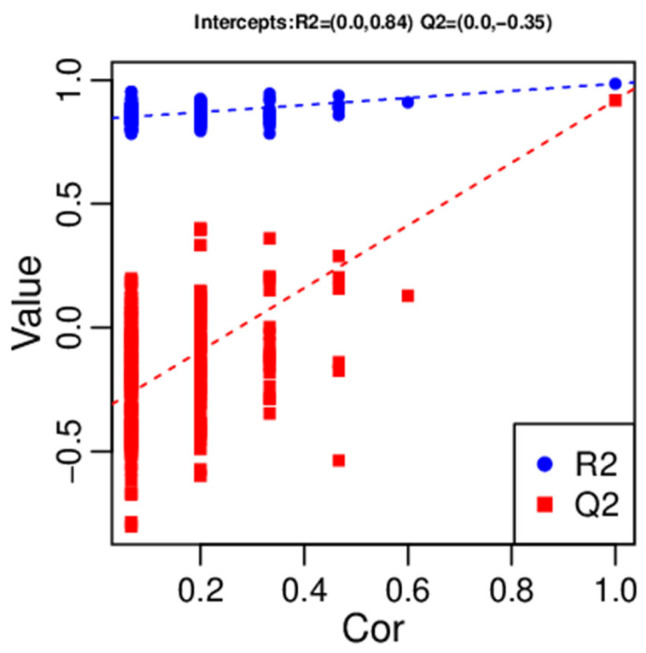
Permutation test of the OPLS-DA model. The *X*-axis represents the correlation between the replacement group and the original model group and the *Y*-axis represents the value of R^2^Y or Q^2^Y (where Q^2^Y and R^2^Y of 1 are the values of the original model). The two points in the upper right corner represent R^2^ and Q^2^ of the actual model the blue and red point to the left represent the R^2^Y and Q^2^Y of the model after replacement, respectively. The two dotted lines represent the regression line fitted by R^2^Y and Q^2^Y.

**Figure 6 ijms-26-09107-f006:**
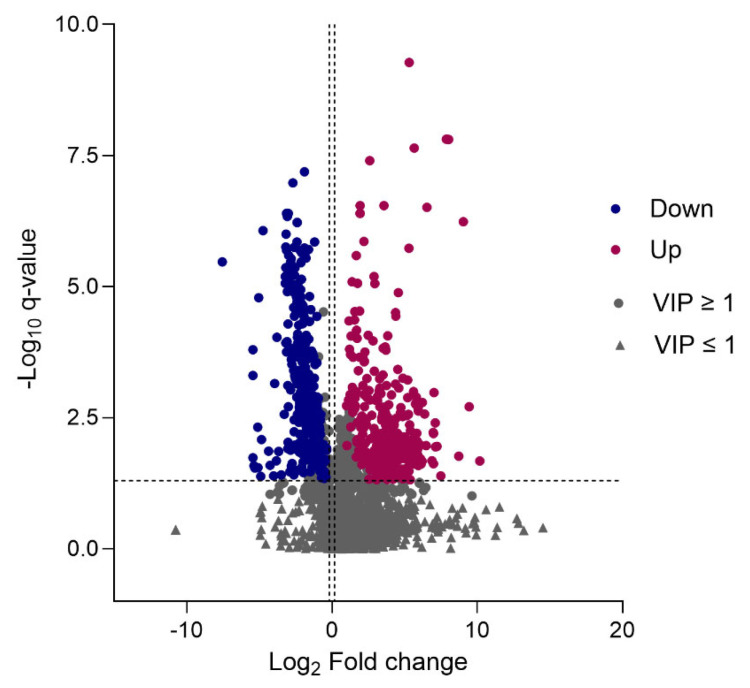
Volcano plot of identified metabolites. Each data point represents an identified metabolite, with up-regulated, down-regulated and insignificant metabolites shown as red, blue and grey points, respectively.

**Figure 7 ijms-26-09107-f007:**
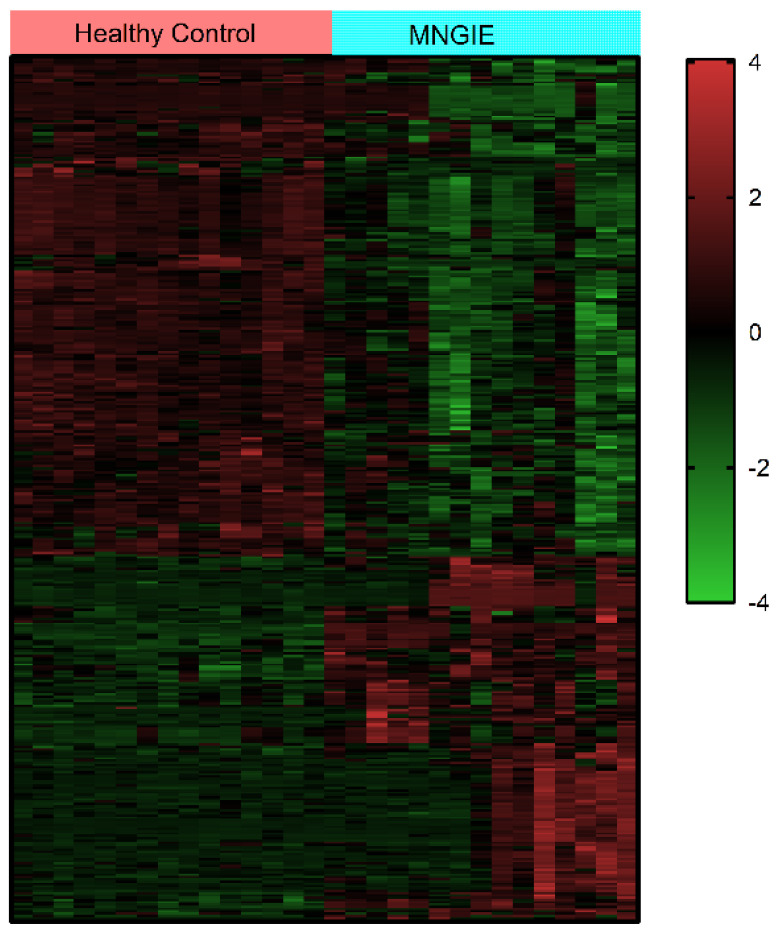
Cluster analysis of the expression levels of the 379 significantly altered plasma metabolites in healthy controls versus patients with MNGIE. Data were log_2_ transformed and z-score normalized. The clustering algorithm used Hierarchical Cluster, and the distance calculation uses Euclidean distance. Each row represents a differential metabolite, and each column represents a sample. The colour represents the expression level, where green to red corresponds to the expression level from low to high.

**Figure 8 ijms-26-09107-f008:**
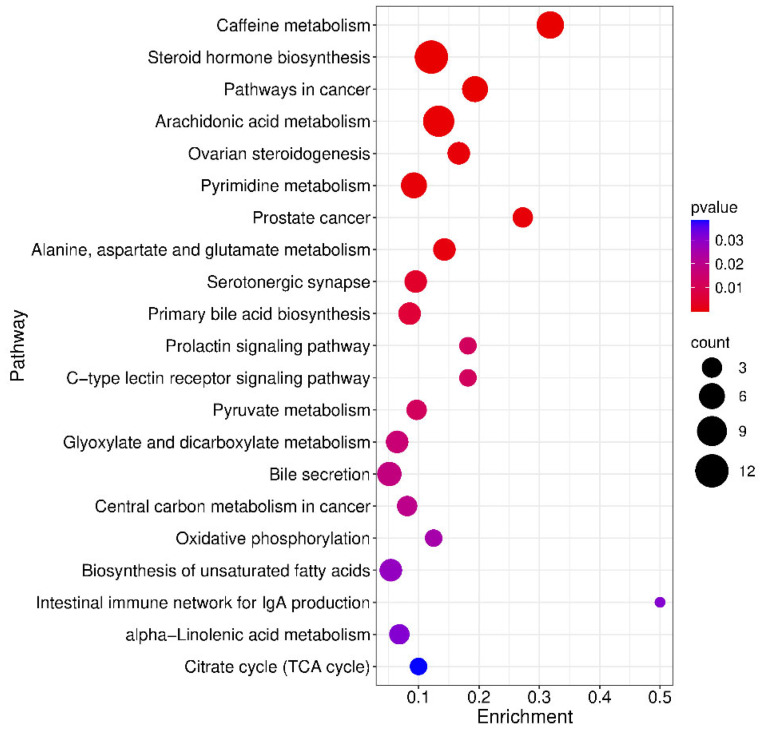
Bubble diagram of the metabolic pathway enrichment analysis. The *X*-axis enrichment factor is the number of identified differential metabolites annotated to this pathway divided by all metabolites known to be annotated to this pathway. The colour and size of each circle is based on the *p*-value and the number of differentially expressed metabolites annotated to each pathway, respectively.

**Figure 9 ijms-26-09107-f009:**
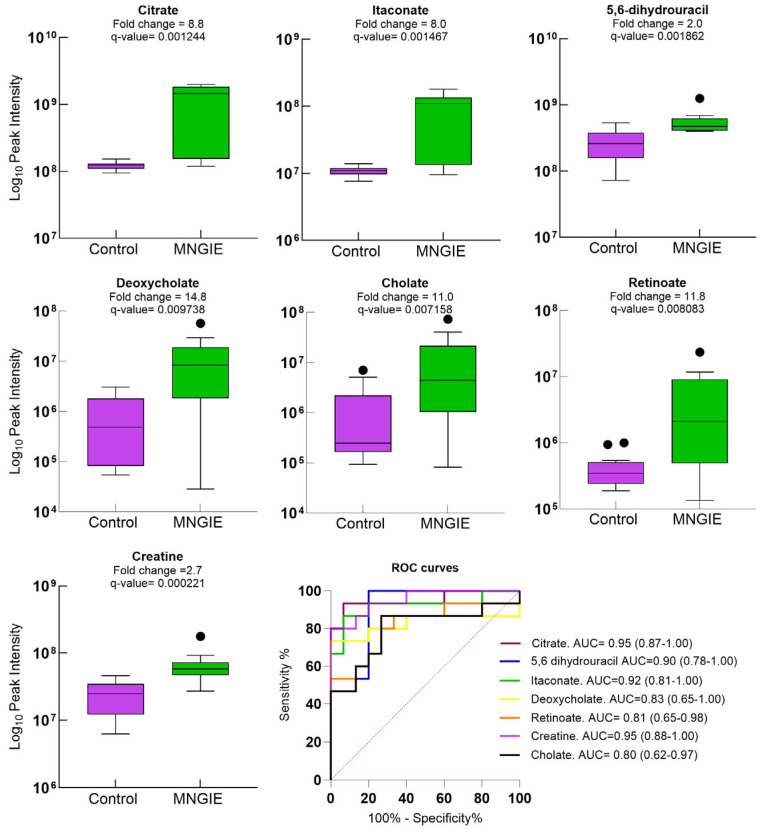
Box and whisker plots of healthy controls and patients with MNGIE for metabolites with a credibility level of 1, *q*-value  <  0.05, fold change  ≥  1.2 and VIP  ≥  1.

**Figure 10 ijms-26-09107-f010:**
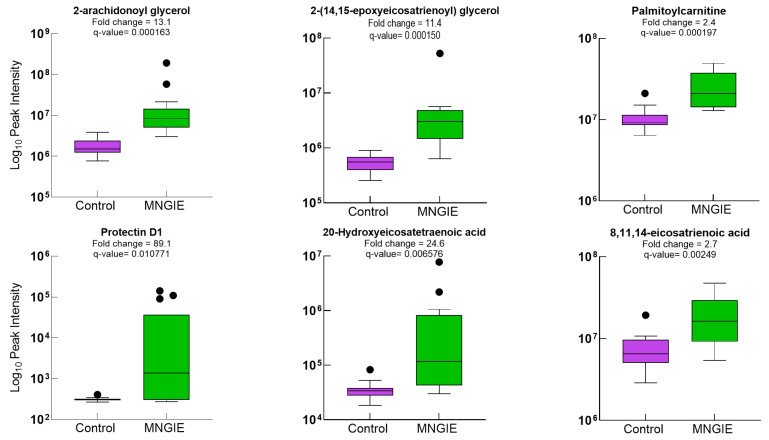
Box and whisker plots of healthy controls and patients with MNGIE for lipid metabolites with a credibility level of 2, *q*-value  <  0.05, fold change  ≥  1.2 and VIP  ≥  1.

**Figure 11 ijms-26-09107-f011:**
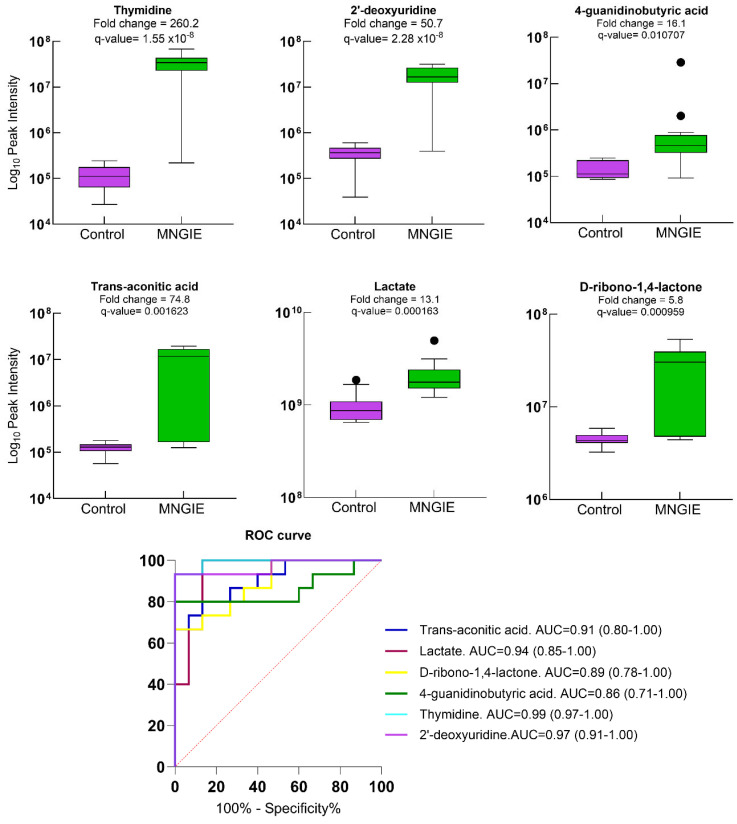
Box and whisker plots of healthy controls and patients with MNGIE for other metabolites with a credibility level of 2, *q*-value  <  0.05, fold change  ≥  1.2 and VIP  ≥  1.

**Figure 12 ijms-26-09107-f012:**
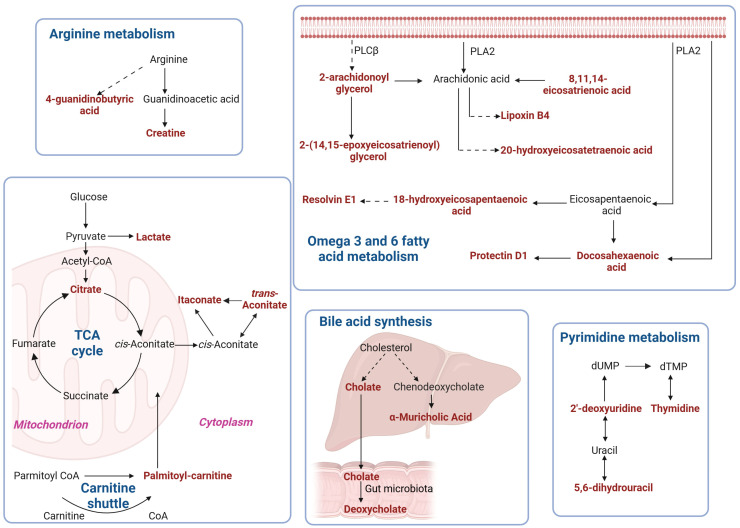
Schematic overview of pathways in which putative plasma biomarkers (labelled in red) were elevated in patients with MNGIE compared to healthy controls. Biomarkers were identified based on *q*-value  <  0.05, fold change  ≥  1.2, VIP  ≥  1, AUC between 0.80 and 1.0, and confidence Levels 1–2.

**Figure 13 ijms-26-09107-f013:**
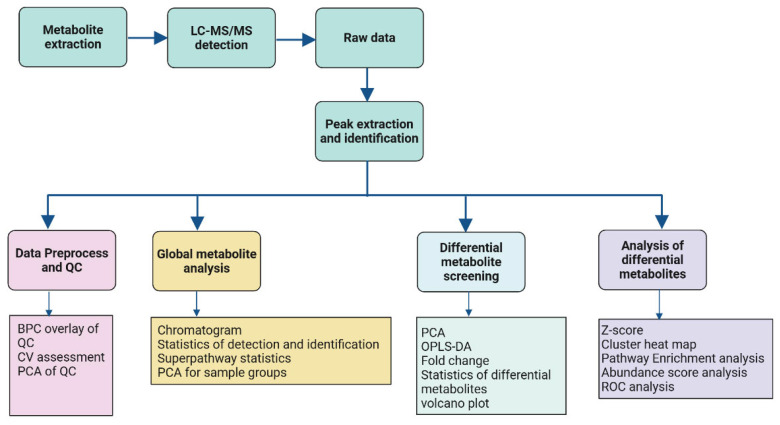
Plasma metabolomics profiling workflow.

**Table 1 ijms-26-09107-t001:** Differential abundance score analysis and pathway regulation status following KEGG pathway enrichment of differential metabolites.

KEGG Pathway	*p* Value	KEGG Names	KEGG Identification	Differential Abundance Score and Pathway Regulation Status
Caffeine metabolism map 00232	2.922 × 10^−8^	7-methylxanthine; 3-methylxanthine; Theobromine; Paraxanthine; Theophylline; 1-Methyluric acid; 1,3,7-trimethyluric acid	C16353 + C16357 + C07480 + C13747 + C07130 + C16359 + C16361	−1 Down
Steroid hormone biosynthesis Map 00140	4.579 × 10^−8^	5α-pregnan-3,20-dione; 17α-hydroxyprogesterone; Androsterone; Androsterone glucuronide; Dehydroepiandrosterone; Aldosterone; Etiocholanolone; Estrone; 20-oxopregn-5-en-3-yl hydrogen sulfate; Tetrahydrocortisol; Androstenedione	C03681 + C01176 + C00523 + C11135 + C01227 + C01780 + C04373 + C00468 + C18044 + C05472 + C00280	−0.8 Down
Pathways in cancer map 05200	7.263 × 10^−6^	Retinoate; Dehydroepiandrosterone (dhea); Prostaglandin e2; Fumaric acid; Androstanolone; Androstenedione	C00777 + C01227 + C00584 + C00122 + C03917 + C00280	0 Down
Arachidonic acid metabolism Map 00590	1.679 × 10^−4^	Lipoxin b4; 11-dehydro thromboxane b2; Thromboxane b2; Prostaglandin e2; Prostaglandin a2; 20-hydroxy-(5z,8z,11z,14z)-eicosatetraenoic acid; 15-keto prostaglandin f2α, 11,12-DHET, 15-(S)HPETE, 11,12-EET	C06315 + C05964 + C05963 + C00584 + C05953 + C14748 + C05960 + C14774 + C05966 + C14770	0.8 Up
Ovarian steroidogenesis Map 04913	4.820 × 10^−4^	17α-hydroxyprogesterone; Dehydroepiandrosterone (dhea); Estrone; Androstenedione, 11,12-EET	C01176 + C01227 + C00468 + C00280 + C14770	−1 Down
Pyrimidine metabolism map 00240	5.228 × 10^−4^	L-glutamine; 5,6-dihydrouracil; Uracil; 2′-deoxyuridine; Thymine; Thymidine	C00064 + C00429 + C00106 + C00526 + C00178 + C00214	0.7 Up
Prostate cancer map 05215	5.652 × 10^−4^	Dehydroepiandrosterone (dhea); Androstanolone; Androstenedione	C01227 + C03917 + C00280	−1 Down
Alanine, aspartate and glutamate metabolism map 00250	8.844 × 10^−4^	Citrate; L-glutamine; Fumaric acid; 2-keto-glutaramic acid	C00158 + C00064 + C00122 + C00940	0 Down
Serotonergic synapse map 04726	4.076 × 10^−3^	11-dehydro thromboxane b2; Thromboxane b2; Prostaglandin e2; Prostaglandin a2, 11,12-EET	C05964 + C05963 + C00584 + C05953 + C14770	1 Up
Prolactin signaling pathway map04917	0.012158	Estrone; Androstenedione	C00468 + C00280	−1 Down
C-type lectin receptor signaling pathway map 04625	0.012158	Prostaglandin e2; Fucose	C00584 + C01019	1 Up
Pyruvate metabolism map 00620	0.012235	Fumaric acid; S-lactoylglutathione; 2-butynedioic acid	C00122 + C03451 + C03248	0.3 Up
Glyoxylate and dicarboxylate metabolism map 00630	0.015951	Citrate; L-glutamine; 4-hydroxy-2-oxoglutaric acid; 3-oxalomalic acid	C00158 + C00064 + C05946 + C01990	−0.5 Down
Bile secretion map 04976	0.017976	Cholate; Deoxycholate; Thromboxane b2; Prostaglandin e2; Bilirubin	C00695 + C04483 + C05963 + C00584 + C00486	0.6 Up
Central carbon metabolism in cancer map 05230	0.019754	Citrate; L-glutamine; Fumaric acid	C00158 + C00064 + C00122	0.3 Up
Oxidative phosphorylation Ma p00190	0.025202	Fumaric acid; Ubiquinol 10	C00122 + C11378	0 Down
Biosynthesis of unsaturated fatty acids ma p01040	0.028497	Docosahexaenoic acid; Dihomo-gamma-linolenate; Eicosapentaenoate; Nervonic acid	C06429 + C03242 + C06428 + C08323	0.8 Up
Intestinal immune network for IgA production map 04672	0.030961	Retinoate	C00777	1 Up
alpha-Linolenic acid metabolism map00592	0.031107	13(s)-hotre; 12-oxo phytodienoic acid; 9(s)-hotre	C16316 + C01226 + C16326	0.3 Up
Citrate cycle (TCA cycle) map 00020	0.038311	Citrate; Fumaric acid	C00158 + C00122	0.5 Up

## Data Availability

The raw data are available from the corresponding author on reasonable request.

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
