# Peer review of "Untargeted Plasma Metabolomics Extends the Biomarker Profile of Mitochondrial Neurogastrointestinal Encephalomyopathy"

_ijms, 2025, doi:10.3390/ijms26189107_

Round 1
Reviewer 1 Report
Comments and Suggestions for Authors
The manuscript under review addresses a pertinent research area in the field of untargeted metabolomics, focusing on the comparative analysis between patients and healthy controls. The topic presents clear scientific interest due to the underexplored nature of metabolic biomarkers in mitochondrial neurogastrointestinal encephalomyopathy. However, the study exhibits significant methodological and reporting deficiencies that compromise the robustness of the proposed conclusions.
Critical Methodological Limitations
- Study Design and Sample Size
The experimental cohort comprises only 15 individuals per group, which constitutes a fundamental limitation affecting both the statistical power of the analysis and the generalizability of the obtained results.
- Analytical Method
The liquid chromatography-mass spectrometry (LC-MS) methodology was selected for non-polar and medium-polar metabolites. This configuration proves inadequate for efficient retention of highly polar compounds, despite several of these metabolites being discussed as significant in the analysis.
A considerable number of compounds (>20 according to the supplementary table) elute at retention times between 0.5-0.8 minutes, coinciding with or approaching the system's dead time. These compounds cannot be considered analytically reliable and require explicit justification for their interpretation. In addition, ittaconate and other key metabolites elute in the front.
- Extraction Procedure
The employed extraction method presents known limitations in the recovery of lipid species. This limitation must be explicitly stated and considered in the interpretation of lipid-related results
- Quality Control and Quality Assurance
It is necessary to provide detailed information regarding:
Quality control (QC) sample preparation procedures and confirmation of identical treatment between QC samples and study samples
Number of system equilibration injections
Sample randomization protocol
The QC1-QC5 samples referenced in the supplementary Excel file are not available. Additionally, Figure 2 shows QC samples separated from study samples, a situation requiring explanation.
- Internal Standards
Clarification is required regarding the application of internal standards (IS), given that the manuscript indicates normalization through Z-scores, suggesting that IS were not utilized. Most employed IS correspond to polar compounds that could elute at dead time, compromising their utility for quality control.
- Metabolite Annotation
It is essential to provide:
Detailed annotation criteria
Confirmation of consideration of isotopic patterns, neutral losses, and in-source fragmentation
Information on experimental mass, exact mass, ppm error, and experimental adduct type
Review of annotations with mass errors in the order of thousands of ppm (suppl. table)
A clear definition of the term "match" used in assignments
Specification of which of the 23 "identified" metabolites were confirmed with authentic standards
Data Presentation Deficiencies
Figures
Figure 1: The right-hand legend is illegible
Figure 3: The metabolite classification lacks reference or justification
Figure 4: The PC2 value reported as 99% appears implausible
Figure 7: Image quality hinders interpretation
Figures 9-11: All metabolites appear increased in patients, a pattern requiring discussion of possible biases
Statistical and Pathway Analysis
The fold-change thresholds (≥1.2 or ≤0.83) are asymmetric without apparent justification. The pathway analysis highlights caffeine metabolism as the most enriched pathway, a result of questionable biological relevance that requires exclusion or justification.
Formatting Considerations
It is recommended to replace "UPLC" with "UHPLC" throughout the text to maintain current standard terminology.
Despite addressing a clinically important topic, the manuscript in its current form presents methodological and presentation limitations that prevent reliable evaluation of its scientific contributions. The suggested revisions are essential to ensure the robustness and reproducibility of the reported findings.
Reviewer 2 Report
Comments and Suggestions for Authors
See the attached Files
